# Courage, Career Adaptability, and Readiness as Resources to Improve Well-Being during the University-to-Work Transition in Italy

**DOI:** 10.3390/ijerph18062919

**Published:** 2021-03-12

**Authors:** Paola Magnano, Ernesto Lodi, Andrea Zammitti, Patrizia Patrizi

**Affiliations:** 1Faculty of Human and Social Sciences, Kore University, Cittadella Universitaria, 94100 Enna, Italy; 2Department of Humanities and Social Sciences, University of Sassari, Via Roma, 151, 07100 Sassari, Italy; elodi@uniss.it (E.L.); patrizi@uniss.it (P.P.); 3Department of Science of Education, University of Catania, Via Biblioteca 4, 95124 Catania, Italy; andrea.zammitti@phd.unict.it

**Keywords:** flourishing, courage, career adaptability, career transition readiness, college students, life satisfaction, career transition

## Abstract

College students approaching a university degree can experience a critical period in their career development path that could affect their well-being. The main aim of this study was to examine the role of courage, career adaptability, and professional readiness as protective factors toward life satisfaction and flourishing during the university-to-work transition. These psychosocial resources could be useful to cope with the recent transformations of the labor market. The study involved 352 Italian university students (M = 100; F = 252), aged from 21 to 29 years (M = 23.57; SD = 2.37), attending the last year of their degree course. The results of the mediation analysis showed that courage plays a mediating role between career transition readiness and career adaptability, on one hand, with well-being indicators as outcomes. The results are discussed, providing some suggestions on practical implications for career interventions to support college students during the university-to-work transition.

## 1. Introduction

In the first decades of the 21st century, significant changes have occurred with respect to the conception of work and the development of a professional self within the paths of one’s life and career trajectories. The current context characterized by rapid social and technological changes and by phenomena that increase risky situations (for example, globalization, unemployment, sense of insecurity and precariousness, social inequality) requires a more malleable, complex, and nonlinear career path development process [1,2]. This process seems to involve, especially in career transition periods, many psychological resources to face the growing perception of negativity about the future, which is frequently characterized by feelings of discomfort and hopelessness [3]. The concept of a “boundaryless career” [4,5] reflects the idea of the change that the concept of career has undergone in the last twenty years.

The university-to-work transition literature has explored the influence of sociodemographic variables (gender, nationality, type of school), personality (open-mindedness, locus of control), and career development variables (decision-making and career planning) [6,7]. Moreover, a large body of literature has shown that additional psychological resources are involved in dealing effectively with career transitions: readiness, defined as the motivation to make a career transition, and career adaptability, which is “a psychosocial construct that denotes an individual’s readiness and resources for coping with current and imminent vocational development tasks, occupational transitions, and personal traumas” [8] (p. 51). The transitions, although no longer the only “elective territory” for career counseling and career education interventions, nevertheless represent the critical steps to increase feelings of uncertainty and the experience of concern and discomfort for individuals. Savickas [6] identified unpredictability as the prominent characteristic of career paths in the third millennium; according to the author, the instability that characterizes the world of work requires individuals to be able to respond adaptively to continuous changes: A successful career transition can depend on individuals’ ability to manage the different contextual conditions related to the transition and on the full use of their psychological resources [9]. In Savickas’ theories [8,10], successful career development is a continuous process of adaptation resulting from person-environment integration, where people have to actively build their careers, responding effectively to the environmental challenges through acquiring career resources [11]. The Life Design paradigm underlines the relevance of complex dynamics and multiple nonlinear interactions involved in the evolution of people’s life and career trajectories. Therefore, the Life Design approach provides theoretical and methodological responses to career counselors who operate in an unpredictable context regarding the future possibilities, given that individual, educational and work environments tend to change suddenly, and can be fluid and unstable [3,12,13,14]. Characteristics such as career adaptability, career transition readiness, and courage emerge in this framework as positive psychological resources allowing young persons to manage their career paths, negotiate transitions and design their futures. They can be considered career resources that can help young professionals to respond more efficiently given the current context of fluidity and risk in building a successful and satisfactory career path. The use of these resources will probably determine the future successful careers of recent graduates [15,16,17], enabling them to successfully face the university-to-work transition and future transitions, ensuring their general and domain-specific well-being.

We briefly introduce the variables of interest in our study in a more analytical way. As mentioned above, a crucial variable in the Life Design paradigm is career adaptability: It concerns the management of professional tasks, role transitions, and coping strategies that people use to deal with these changes; it is the process of actively building one’s own professional life by facing changes and considering the social context [8]. Career adaptability is essential to help individuals in planning for their uncertain future, facing unfavorable working conditions, adapting to changes in labor market conditions, and finally increasing their well-being [12].

Similarly, career transition readiness represents “the extent to which one is task-oriented and motivated to move ahead with the career transition” [18] (p. 65); this construct reflects how individuals appraise their motivation for making a career transition. Individuals who have more motivation tend to be in the career change process for a shorter period and feel they are making more progress in their change [18].

Among the positive resources, Peterson and Seligman [19] identified courage as one of six dimensions of human strengths that fosters well-being at both the individual and community levels. In particular, Seligman [20] put courage among the 24 strengths and virtues that help people to flourish. Human virtues and strengths, according to some authors [21], allow individuals to create a “good life,” and courage is necessary because, despite humans’ vulnerability, it enables people to protect something considered important, through adequate management of physical, psychological, or social risks. Therefore, if courage responds to these characteristics, our future, especially in periods of transition and insecurity, is something important to protect, despite the risks it entails. Courage has been found to play the role of a protective factor to face risky situations, to cope with stressful conditions, to make career choices, and to plan career paths despite current fears [22]. It has the power to reduce negative emotions, such as discomfort and apprehension, that can negatively affect life satisfaction [23,24] and well-being [25]. Ginevra et al. [26] stated that courage can be defined as “an adaptive behavior to cope with career development tasks and changing work and career conditions and for promoting life satisfaction” [26] (p. 459).

These career-related individual resources can simultaneously perform a protective function for the success and well-being experienced in career paths, especially during transitions and given the current context, presupposing a proactive and constructive function toward individuals’ own career projects and effective response to contextual challenges. However, they involve different nuances: Courage has a more behavioral dimension, presuming acting in a courageous manner rather than a perception of risk; career readiness presupposes a more motivational aspect and dealing and coping with perceived environmental barriers, especially in a period of transition; career adaptability includes cognitive and socio-cognitive factors of beliefs and attitudes in the exploration of oneself and the context, i.e., the capacity to adapt and integrate different individual and contextual factors in a long time and space with respect to career readiness. These protective factors can also support people in coping with the probable career shocks that most people encounter today, that is, disruptive and extraordinary events caused by factors beyond the control of the individual, which significantly affect the career development process, having a direct impact especially on young professionals [6,27,28].

There are two main reasons for the choice of two well-being outcomes in our study: (a) due to the growing attention dedicated to promoting well-being by all psychological disciplines, related to the contribution of the development of positive psychology; and (b) because the well-being experienced at university has demonstrated important effects not only on how students approach career transitions but also on some variables for new graduates related to the world of work.

As for the first reason, in the positive psychology framework, attention to the well-being and quality of life of people in various life contexts is increasing. Of the two approaches most studied in this area, i.e., the hedonistic and eudaemonic perspectives, a complex vision of well-being emerges that can be seen as linked to both positive emotions and life satisfaction and a state of tension toward growth and self-realization. Seligman [20] also implemented his vision of well-being, moving from the theory of authentic happiness, where the most important criterion of measurement was life satisfaction, to the theory of well-being, in which the criterion of measurement became flourishing. Flourishing refers to individuals’ perception of being able to fully realize their potential, a condition that leads to complete prosperity, which is the realization of their own qualities within the social dimensions. It is a range of positive internal experiences toward not only individual self-realization but also positive functioning in the relationship with the world.

As for the second reason, over the past 10 years, the literature on career development has highlighted how satisfaction with school and university choices (derived from opportunities to use one’s skills, develop one’s interests, and implement one’s self-concept) is linked in young people to the use of positive resources and planning skills in career projects and other non-intellective competencies [29,30,31,32,33,34,35]. The level of satisfaction in the university phase can provide information on future positive dimensions related to work. In particular, Blanch and Aluja [36] stated that satisfaction with university courses is predictive of subsequent involvement in work, and Lo Presti and colleagues [37] showed in a sample of new graduates that academic satisfaction was a mediator between career competencies and subjective career success through employability activities.

Using positive psychology as an additional framework, in this study, the evaluation of well-being is conducted not only through a global measure of satisfaction with life but also using the concept of flourishing [38], because Seligman [20] stated that flourishing enhances the chance to have successful and rewarding careers. People in flourishing conditions have higher levels of happiness and satisfaction; they see their life as meaningful, knowing how to relate constructively and independently with the environment; they experience feelings of personal growth and have more control over life events [39]. Moreover, some studies have found a positive relationship of flourishing level with academic performance, self-control, and the use of mastery-approach goals and, on the other hand, a negative relationship with academic procrastination [40,41]. Finally, Van Zyl and Stander [42] suggested improving students’ flourishing (and the skills necessary to get it) early in their academic careers.

Finally, career adaptability, career transition readiness, and courage in university students who are approaching the university-to-work transition, can play a role as protective factors for their perceived well-being in the transition’s critical phase.

## 2. Literature Review: The Role of Career Adaptability, Career Transition Readiness, and Courage in Career Path Development and as Well-Being Indicators

### 2.1. Career Adaptability

Career adaptability is composed of four factors: (a) concern, related to the time perspective and personal commitment to the future, viewing it with an optimistic and hopeful vision; (b) control, referring to the individual’s perception of being able to implement an adequate decision-making process with respect to one’s own career; (c) curiosity, related to the propensity for curiosity and exploration, which is considered relevant because knowledge derived from curiosity and exploration of possibilities is useful for making choices; and (d) confidence, theorized as the anticipation of success in facing challenges and overcoming obstacles to pursue career-related goals [43].

People with high levels of career adaptability also have higher levels of internal locus of control, optimism, persistence, self-efficacy beliefs, coping and problem-solving skills, ability to search for information, career decidedness, responsibility and commitment to career choices and career decision-making, ability to imagine multiple possible future scenarios, open-mindedness and search for opportunities, academic performance, and well-being and a lower possibility of perceiving internal and external career barriers [34,35,36,37,38,39,40,41,42,43,44,45,46,47,48,49,50]. Hirschi [29] showed that levels of adaptability can predict the level of control in one’s life and one’s perceived well-being. A meta-analytic study [51] showed that the relationship between career adaptability and professional well-being (academic, school, work, and career satisfaction) was moderated by the career stage, among other variables. Regarding career transitions, longitudinal studies have shown that career adaptability facilitates young people in successfully managing their career transition, especially people with higher levels of confidence, curiosity and exploration, decision-making and planning [2,52,53,54,55]. Career adaptability increases the possibility of reemployment and long-term career success [7,56], and it is positively related to both general and professional well-being [57]. A study conducted by Konstam, Celen-Demirtas, Tomek, and Sweeney [58] showed that in an unemployment situation, which is often characterized by anxiety and depression, career adaptability is a protective factor of subjective well-being: Emerging adults who showed higher levels in the subdimensions of control and confidence also revealed higher levels of life satisfaction. Moreover, career adaptability, in a recent meta-analytic study [59], was positively correlated with many career, work, and subjective well-being outcomes, for example, with a disposition to be flexible and able to make career transitions and to behave adaptively in exploring and planning career paths, increasing career satisfaction, improving the chance to have an effective organizational commitment, and decreasing the possibility to perceive job stress. A recent study [60] showed a positive effect of career adaptability on job and life satisfaction and a negative effect on perceived stress levels in life, demonstrating that career adaptability can affect long-term individual adaptation due to its contribution to maintaining levels of well-being. There is a positive relationship between career preparation and positive youth development, not only considering the effect on successful career transitions but also as a crucial component strictly related to well-being and a decrease in perceived stress [30,61,62].

Finally, career adaptability can facilitate university adjustment, promoting well-being in the school-to-college transition [63], and, also in young adults and old adults in career transition, it is associated with various indicators of subjective well-being [64]. Therefore, Cabras and Mondo [65] found that university students with a higher level of career adaptability were more satisfied with life through a mediation role of the future perspective. Conversely, there are not many empirical studies on the relationship between career adaptability and flourishing. Some studies have shown career adaptability to influence flourishing, partially mediated by the presence of life meaning [66].

### 2.2. Career Transition Readiness

Career transitions have now become a standard in the professional world; undoubtedly, the transition from university to work can be considered a fundamental step for graduate students [15]. The watchword of the new century is “flexibility” [67], which needs to be managed through various personal resources [16] to be successful professionally. However, the way the individuals react is not always the same and depends on their resources and perceived barriers [68]. Readiness is among the fundamental components of psychological resources to effectively deal with career transitions. In Heppner’s [69] conception, readiness is defined by how much individuals are willing to exert themselves to reach their career goals; it indicates how individuals are motivated to make a transition that affects their professional future [68]. People at different levels are prepared for change and differ in their willingness or readiness to influence change [17]. Previous studies [70,71] have shown that readiness is related to the ability to seize opportunities and face obstacles that may be encountered in career choice. Furthermore, a proactive attitude toward career-building positively influences career identity and is a mediator in the relationship between satisfaction and career identity [29]. Therefore, readiness makes it easier for students to identify with their career during the university course and facilitates the transition from university to work; this allows the students to acquire the ability to process feedback on themselves [72] and better interact with the complexity of the world of work [29]. Finally, readiness is positively associated with satisfaction [73] and optimism [74,75] and negatively associated with psychological distress [76]. Some studies specifically focused on career transition readiness in college students showed that career adaptability was among its predictors [77] and there are positive correlations between, career readiness, career adaptability, and satisfaction with life [78]. In the university context, it seems that one of the most important factors associated with the decrease in self-efficacy in a career search and the increase in psychological distress is the reduction of certain psychological resources such as readiness, confidence, and support in a career transition [79].

### 2.3. Courage

The recent literature on courage agrees that at the basis of the construct of courage can be recognized the presence of subjective perception of fear [80]. Therefore, one of the most used definitions of courage derives from Norton and Weiss [81]: Courage involves the ability to persist and perpetuate efforts despite a subjective feeling of fear. According to some authors, the ability to act courageously with respect to fear is reinforced by other personal characteristics, some deriving from positive psychology, such as, for example, hope, optimism, resilience, and others related to personal beliefs (i.e., values). Some authors also underline the component of risk in the courage definition; for example, Rate, Clarke, Lindsay, and Sternberg [82] introduced coping with risk and obstacles as a component of their model of courageous behaviors. Many studies in recent years have focused on the relationships between courage and career-related variables because the tendency to behave courageously, despite fears and perceived risks, can influence how people succeed or not in facing new challenges. Courage seems to affect both educational and work positive results, for example, educational and professional well-being outcomes [83,84,85]. In this field, courage has been found to be positively associated with career adaptability, positive working behavioral outcomes, prosocial behavior, self-efficacy, leadership and the ability to achieve long-term objectives, positive affect, emotional intelligence, openness, extraversion, resilience, and motivation to achieve aims by implementing alternative solutions; people with higher levels of courage feel less fear and show more coping skills [2,86,87,88,89,90,91,92,93]. Some studies have demonstrated that courage is related to psychological capital (a construct composed of confidence, resilience, hope, and optimism), influencing domain-specific and overall life satisfaction, flourishing and other well-being indicators, such as psychological and subjective well-being [2,94,95,96,97,98,99]. Putman [100] stated that psychological courage is crucial to cope successfully with negative habits and irrational anxieties, and its development is a key factor to improve personal well-being. Consequently, individuals with higher levels of courage tend to show better personal well-being [101].

Finally, it should be noted that the study and promotion of courage increasingly take place in childhood [102,103], also as a fundamental phase for the construction of children’s career aspirations. Many studies have shown that courage plays a fundamental mediation role between career-related predictors toward positive outcomes in the field of career construction and educational and professional well-being [2,98,104].

## 3. Aims of the Study

Based on the literature review, career adaptability and readiness can be considered core psychological constructs related to subjective well-being in the university-to-work transition [8,69]; moreover, courage has been found to have a role in coping with career development tasks, promoting life satisfaction [26].

Thus, the study presented aimed to verify the role that courage plays in the relationship between readiness and the four dimensions of career adaptability (concern, control, curiosity, and confidence) and two indicators of subjective well-being, satisfaction with life, and flourishing. More specifically, we hypothesized the following:

**Hypothesis 1a** **(H1a).**
*Courage mediates the association between readiness and life satisfaction.*


**Hypothesis 1b** **(H1b).**
*Courage mediates the association between readiness and flourishing.*


**Hypothesis 2a** **(H2a).**
*Courage mediates the association between concern and life satisfaction.*


**Hypothesis 2b** **(H2b).**
*Courage mediates the association between concern and flourishing.*


**Hypothesis 3a** **(H3a).**
*Courage mediates the association between control and life satisfaction.*


**Hypothesis 3b** **(H3b).**
*Courage mediates the association between control and flourishing.*


**Hypothesis 4a** **(H4a).**
*Courage mediates the association between curiosity and life satisfaction.*


**Hypothesis 4b** **(H4b).**
*Courage mediates the association between curiosity and flourishing.*


**Hypothesis 5a** **(H5a).**
*Courage mediates the association between confidence and life satisfaction.*


**Hypothesis 5b** **(H5b).**
*Courage mediates the association between confidence and flourishing.*


## 4. Materials and Methods

### 4.1. Participants

The study involved 352 university students (M = 100; F = 252), aged from 21 to 29 years (M = 23.57; SD = 2.37), attending the last year of their degree course. According to our sample size calculation, our sample is capable of reflecting the target population for a confidence level of 95% and at a 6.17% confidence interval. The students attended different degree courses: 39.8% attended degree courses in the socio-psychological and educational area; 20.2% attended degree courses in physical and sports activities; 15.6% attended humanistic degree courses; 8.5% attended economic, political, and juridical degree courses; 7.4% attended health profession courses, and; the remaining participants were distributed as follows: tourism, biological and natural sciences, and engineering. The respondents were recruited from different universities in the north, center, and south of Italy, with the aim of obtaining data from different socio-cultural backgrounds. They completed an online survey after having expressed their informed consent to the use of data; they were free to abandon the compilation at any time. The respondents were required to provide a personal code, composed of their birth year, the first two letters of their last name, and the first two letters of their first name. This code, matched with other demographic information, allowed us to check any potential double compilation. The survey was approved by the ethical committees of the universities involved and followed the rules of the Italian Association of Psychology for psychological research.

### 4.2. Measures

#### 4.2.1. Career Adapt-Abilities Scale-Italian Form

The Italian form [48] of the Career Adapt-Abilities Scale-International Form 2.0 [11] is composed of 24 items that measure four subscales: Concern (sample item: Thinking about what my future will be like), Control (sample item: Making decisions by myself), Curiosity (sample item: Looking for opportunities to grow as a person), and Confidence (sample item: Overcoming obstacles). The responses were given on a 5-point Likert scale from 1 (not strong) to 5 (strongest). Cronbach’s alphas of the study sample for the four subscales are 0.87, 0.84, 0.87, and 0.91, respectively.

#### 4.2.2. Readiness Scale

The Readiness scale is a subscale of the Career Transition Inventory [18,105] (Italian adaptation [73]). It measures the perception of the psychological resources available to the individual who is experiencing a career transition. The Readiness subscale is composed of 13 items measuring individuals’ motivation to move forward with the career transition. Participants responded on a 6-point scale ranging from completely disagree to completely agree. A sample item is “Each day I do something on this career transition process, I would say I’m motivated.” Cronbach’s alpha calculated for the study sample is 0.84.

#### 4.2.3. Courage Measure

The Italian version of the Courage Measure (CM; [26,81]) was adapted from the short version of the CM proposed by Howard and Alipour [94]; it is composed of six items rated on a 7-point Likert scale from 1 (never) to 7 (always). A sample item is “I tend to face my fears.” Cronbach’s alpha of the study sample is 0.82.

#### 4.2.4. Satisfaction with Life Scale

The Satisfaction with Life Scale (SWLS; [106]) measures general life satisfaction using five items rated on a 7-point Likert scale (1 = strongly disagree; 7 = strongly agree). We used the Italian version validated by Di Fabio and Gori [107] (Cronbach’s alpha 0.88), which has shown good psychometric features in terms of factor structure, reliability, and validity. A sample item is “In most ways, my life is close to my ideal.” Cronbach’s alpha of the study sample is 0.88.

#### 4.2.5. Flourishing Scale

The Flourishing Scale [108] is a one-dimensional measure of meaning and purpose in life. We used the Italian version validated by Di Fabio [109] (Cronbach’s alpha 0.88), which has shown good psychometric features in terms of factor structure, reliability, and validity. A sample item is “I am engaged and interested in my daily activities.” It is composed of eight items rated on a 7-point Likert scale (1 = strongly disagree; 7 = strongly agree). Cronbach’s alpha of the study sample is 0.88.

### 4.3. Data Analyses

Using Lisrel 8.80, we first tested the fit of the measurement model, running a confirmatory factor analysis (CFA). The goodness-of-fit indexes used were chi-square (χ^2^), the χ^2^/df ratio (which should be <3), the comparative fit index (CFI), the root mean square error of approximation (RMSEA), and the standardized root mean square residual (SRMR). For the CFI, values over 0.90 suggest acceptable fit, whereas values over 0.95 suggest a good fit. According to Browne and Cudeck [110], a RMSEA < 0.09 is still an acceptable threshold in smaller samples; for the SRMR, a value less than 0.08 is generally considered a good fit [111]. Then, we tested the structural equation model using maximum likelihood estimation, evaluating the same fit indexes.

In the third step, we conducted the mediational analysis, reporting the significance of the indirect effects obtained through the bootstrapping method with 5000 repetitions, with a confidence interval (CI) of 95%. Finally, other well-known statistical analyses were conducted using SPSS 25.0 (IBM, Armonk, NY, USA).

To optimize the sample size, missing values for the relevant items were estimated using the expectation-maximization method. None of the items had more than 5% missing values, indicating that this option was appropriate for use [112].

## 5. Results

### 5.1. Descriptive Statistics and Correlations

The relationships between the variables of the study, including age, were analyzed using Pearson’s r coefficient. All the correlations between the psychological dimensions were significant: We found very strong relationships between the four dimensions of career adaptability and courage, which we assumed as the antecedents in our research hypotheses. Moreover, very strong correlations were found between our antecedents and the components of subjective well-being, satisfaction with life, and flourishing. Conversely, the relationships between the psychological dimensions and age were not significant, except for weak correlations between age and curiosity, confidence, and life satisfaction.

Skewness and kurtosis of the distribution of the total scores of the dimensions of the study were calculated: Skewness values were between −0.87 and −0.17; kurtosis values were between −0.43 and 1.27.

Table 1 reports the descriptive statistics and Pearson’s coefficients.

### 5.2. Gender Differences

The comparison between males and females conducted through the Student’s *t*-test (*p* < 0.05) showed no significant differences in the analyzed dimensions. The results are reported in Table 2.

### 5.3. CFA of the Measures

Before testing the structural model, we conducted an assessment of the properties of the measurement model. We ran a CFA, using Lisrel 8.80, according to Harman’s single-factor test to determine the extent to which common-method variance was a problem. We compared the hypothesized model and a model with one factor, in which all the items loaded on a unique factor, demonstrating that the hypothesized model provided a better fit for the data in all the CFA fit measures (8-factor model: χ^2^(1457) = 3800.110; χ^2^/df = 2.60; CFI = 0.96; RMSEA = 0.07; SRMR = 0.08; AIC = 4265.306; 1-factor model: χ^2^(1484) = 7091.425; χ^2^/df = 4.78; CFI = 0.90; RMSEA = 0.14; SRMR = 0.10, AIC = 12,536.361). The differences were significant according to a comparison of the models’ χ^2^ values and degrees of freedom: Δχ^2^(27) = 3291.315 (*p* < 0.001). According to these results, we found no evidence for common-method bias in the data.

All the items loaded significantly on the hypothesized latent variables (standardized factor loadings range 0.22–0.89; *t* > 2.58), indicating convergent validity. All constructs showed excellent values of composite reliability (CR > 0.84) and acceptable values of average variance extracted (AVE > 0.42).

### 5.4. Structural Model

We applied structural equation modeling analysis to test our hypotheses. The main fit indices suggested that the model fit the data adequately, χ^2^(1457) = 3694.322; χ^2^/df = 2.54; CFI = 0.96; RMSEA = 0.07; SRMR = 0.07. Figure 1 presents the final model. All the relationships between the variables are indicated by standardized β. As represented in the figure, readiness is related to both indicators of subjective well-being (satisfaction with life and flourishing) and to courage; among the four dimensions of career adaptability, only concern shows significant relationships with both indicators of subjective well-being; control and confidence are related only to courage; curiosity shows no significant relationships with courage or subjective well-being.

### 5.5. Mediation Analysis

The mediational hypothesis was tested using the bootstrapping method to verify the significance of the indirect effects. Readiness and concern have both a direct effect (DE) and an indirect effect (IE), mediated by courage, on life satisfaction, and only a direct effect on flourishing; control has no direct effect on life satisfaction and flourishing, but the relationship is fully mediated by courage; curiosity and confidence have no direct effect on life satisfaction nor an indirect effect mediated by courage; the relationship between curiosity and flourishing is fully mediated by courage, whereas the relationship between confidence and flourishing is partially mediated by courage.

The mediation results are presented in Table 3, which contains the standardized β, indicating the intensity of the effect, and the 95% CIs, indicating the significance of the effect with a 5% probability of error (CIs that do not contain 0 are significant). The results show that readiness has a direct relationship with life satisfaction (DE = 0.03, CI = 0.017–0.042) and flourishing (DE = 0.04, CI = 0.025–0.051); moreover, the path from courage to life satisfaction is significant, showing a direct and indirect effect of readiness on life satisfaction (IE = 0.01; CI = 0.001–0.009) mediated by courage; no mediational effect was found in the relationship with flourishing (this result confirms Hypothesis 1a, rejecting H1b).

We found a similar pattern in the relationship between concern and the two indicators of subjective well-being. Concern has a direct relationship with life satisfaction (DE = 0.04, CI = 0.004–0.073) and flourishing (DE = 0.04, CI = 0.006–0.066). Furthermore, the path from courage to life satisfaction is significant, showing a direct and indirect effect of concern on life satisfaction (IE = 0.01, CI = 0.003–0.028) mediated by courage; no mediational effect was found in the relationship with flourishing (this result confirms Hypothesis 2a and rejects H2b).

Control and curiosity have no direct effect on life satisfaction (control: DE = 0.03, CI = −0.007–0.075; IE = 0.02, CI = 0.006–0.039; curiosity: DE = 0.017, CI = −0.022–0.057; IE = 0.004, CI = −0.002–0.013) and flourishing (control: DE = 0.04, CI = −0.004–0.076; IE = 0.003, CI = 0.001–0.008; curiosity: DE = −0.01, CI = −0.004–0.026; IE = 0.009, CI = 0.001–0.022); thus, the relationship between control and subjective well-being is fully mediated by courage, confirming Hypothesis 3a and 3b. On the contrary, we could not confirm that courage fully mediates the relationship between curiosity and subjective well-being, as the total effect is not significant (Hypothesis 4a and 4b are rejected). Finally, confidence has no direct effect or indirect effect on life satisfaction; conversely, confidence is significantly related to flourishing, both directly (DE = 0.04, CI = 0.003–0.082) and by the mediation of courage (IE = 0.013, CI = 0.004–0.029) (Hypotheses 5b is confirmed).

## 6. Discussion

The present study aimed to evaluate the role of courage as a mediator between career transition readiness and the four dimensions of career adaptability (concern, control, curiosity, and confidence) with subjective well-being (life satisfaction and flourishing). We hypothesized that all the independent variables would be related to life satisfaction and flourishing through the mediation of courage. Data analysis partially confirmed the research hypotheses.

More specifically, we observed that courage partially mediated the relationship between readiness and life satisfaction. Similarly, previous studies have demonstrated [29,113] that career transition readiness—defined as individuals’ motivation to make a transition that affects their professional future [68]—and life satisfaction are strictly related; courage, through its mediating role, translates into behaviors in this motivation to face transition situations, even under risky conditions, contributing to the perception of a “good life.”

Regarding the mediation of courage in the relationship between the four dimensions of career adaptability and the two indicators of subjective well-being, we found a partial mediation between concern and life satisfaction. The students in this study, who were more concerned about their career (adopting an optimistic and hopeful vision of their future), likely experienced greater satisfaction primarily due to the possibility to act to overcome fear in stressful or threatening situations.

Moreover, courage partially mediates between control and both life satisfaction and flourishing. Thinking that the future is manageable during a career transition and therefore taking responsibility for one’s choices, which is the attitude related to control, can affect one’s ability to think and act courageously, which, in turn, can have an effect on perceived well-being, in terms of both life satisfaction and meaning of life. As previously found in similar studies [2], courage, conceptualized as adaptive behavior, helps students with an internal locus of control, to effectively cope with career development tasks and career conditions perceived as threatening. In addition, a large body of research has related general feelings of control with a sense of meaning in life [114,115,116]. The students involved in this study, who felt more in control of their career, might also derive from this a source of meaning, ultimately leading to greater life satisfaction [117].

Furthermore, a partial mediation of courage was found between confidence and flourishing. As confidence refers to beliefs in one’s own abilities and flourishing refers to the perception of expressing all one’s talents and possibilities, courage enables individuals to overcome their fears and thus perceive that they live a meaningful life.

Finally, there is no mediation between curiosity and subjective well-being. Moreover, it seems interesting to note that according to mediation analyses, curiosity has no direct or indirect effect through courage on subjective well-being. Although curiosity showed significant bivariate correlations with these variables, after accounting for the other adaptability components, these relationships were nonexistent, confirming results presented in previous studies [117]. This could imply that when it comes to well-being, the other components of adaptability are most important. Perhaps the exploration and knowledge of the surrounding world, considering the particular historical moment and during a career transition, may not have a significant impact on dealing with situations of uncertainty and fear during career transitions. That is, more information about the world is perhaps not perceived as sufficient and functional to achieve the full expression of one’s talents and potential and is not able to simultaneously produce well-being on the emotional side to counteract the associated uncertainty about and fear for the future.

Reading our results in the framework of the Life Design paradigm for career construction, they support the positive relationship of career adaptability and career transition readiness with courage, not only for successful career transitions but also related to subjective well-being [29]. The development of career-related psychological resources combined with courageous behavior enhances well-being. As courage has been found to play the role of a protective factor to face various risky and stressful situations, also in career-related tasks [22], it has the power to reduce negative emotions, which can negatively affect life satisfaction [23,24] and well-being [25]. Our results align with other studies conducted using the Life Design paradigm: Ginevra and colleagues [2], for example, found a partial mediating role of courage in the relationship between career adaptability and life satisfaction, using a sample of adolescents. Furthermore, Santisi and colleagues [95] showed the role of courage as a mediator between psychological capital and life satisfaction and flourishing.

Courage, viewed as the capacity to perpetuate efforts despite fear [81], can help people reflect on the consequences of their actions and accept the risk of their actions. In this sense, courage, as adaptive behavior, can sustain students in dealing with the transition from university to work, enabling them to manage uncertainty effectively.

## 7. Limitations

The results of the study presented should be considered in light of its limitations: First, the cross-sectional nature of the study does not allow for reaching predictive conclusions. Second, the convenience sampling unbalanced by gender poses limits to the generalizability of the results to the population. Third, some differences could arise from comparing the students by the degree courses attended. Future research could explore these differences and use a longitudinal design to (1) establish causal relationships between the dimensions studied and (2) monitor the effectiveness of the university-to-work transition for different levels of psychological resources.

## 8. Conclusions

The uncertainty and instability that characterize the world of work today make it necessary for newly graduated students to develop positive resources that allow them to actively adapt to this uncertain environment. Despite the difficulty of predicting career changes in the coming years, students will have to make career choices that will be determinant for their future; the ability to actively adapt to a career is certainly central, both for the first entry into the world of work and for potential future career changes. It will be essential to be able to make sense of the constant changes and integrate the different experiences.

Career counselors and professionals can plan interventions based on the important meanings deriving from the university-to-work transition; our results suggest that career professionals working in university counseling services should propose individual and group interventions to promote subjective well-being and to help students approach the new challenges of the social context. In particular, the interventions could focus on psychological resources (courage, career adaptability, and career transition readiness) to promote university well-being levels and to facilitate well-being in this critical transition.

## Figures and Tables

**Figure 1 ijerph-18-02919-f001:**
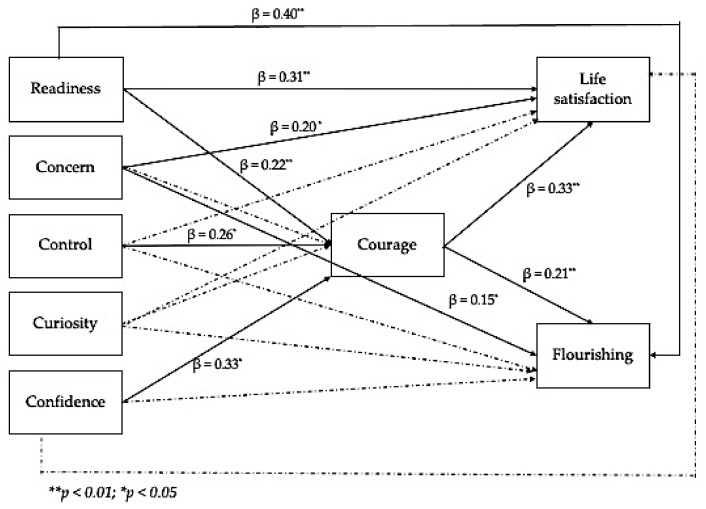
The structural model. Note. Dashed lines indicate nonsignificant relationships.

**Table 1 ijerph-18-02919-t001:** Descriptive statistics and correlations between the dimensions of the study (Pearson’s r).

	M	SD	1	2	3	4	5	6	7	8	9
1. Readiness	55.88	9.89	1								
2. Concern	24.95	3.90	0.36 ***	1							
3. Control	25.89	3.54	0.37 ***	0.66 ***	1						
4. Curiosity	25.22	3.85	0.39 ***	0.62 ***	0.69 ***	1					
5. Confidence	25.90	3.80	0.31 ***	0.64 ***	0.71 ***	0.76 ***	1				
6. Courage	30.91	6.62	0.36 ***	0.44 ***	0.49 ***	0.45 ***	0.50 ***	1			
7. Life satisfaction	24.20	6.33	0.47 ***	0.42 ***	0.42 ***	0.40 ***	0.35 ***	0.47 ***	1		
8. Flourishing	43.97	7.81	0.57 ***	0.52 ***	0.54 ***	0.49 ***	0.52 ***	0.51 ***	0.73 ***	1	
9. Age	23.57	2.37	0.10	−0.01	0.08	0.11 *	0.16 *	0.001	−0.17 *	−0.07	1

*** *p* < 0.001; * *p* < 0.05.

**Table 2 ijerph-18-02919-t002:** Gender differences in the dimensions of the study (Student’s *t* test, *p* < 0.05).

	M (*n* = 100)		F (*n* = 252)		*t*	*p*
	M	SD	M	SD		
1. Readiness	57.12	10.18	55.39	9.74	1.49	0.14
2. Concern	24.39	4.52	25.17	3.61	−1.69	0.09
3. Control	25.85	3.68	25.91	3.48	−0.14	0.89
4. Curiosity	24.97	4.10	25.33	3.74	−0.78	0.44
5. Confidence	25.82	4.13	25.93	3.66	−0.23	0.82
6. Courage	31.68	7.30	30.60	6.32	1.38	0.17
7. Life satisfaction	24.87	6.19	23.94	6.37	1.25	0.21
8. Flourishing	45.17	7.79	43.49	7.78	1.83	0.07

**Table 3 ijerph-18-02919-t003:** Effects of readiness and the four dimensions of career adaptability on life satisfaction and flourishing through courage (standardized β).

Paths	Indirect Effect (IE)		Direct Effect (DE)		Total Effect	
	β	CI 95%	β	CI 95%	β	CI 95%
Readiness–Courage–Life satisfaction	0.005	0.001–0.009	0.028	0.017–0.042	0.033	0.023–0.045
Concern–Courage–Life satisfaction	0.013	0.003–0.028	0.038	0.004–0.073	0.044	0.011–0.080
Control–Courage–Life satisfaction	0.019	0.006–0.039	0.033	−0.007–0.075	0.045	0.004–0.084
Curiosity–Courage–Life satisfaction	0.004	−0.002–0.013	0.017	−0.022–0.057	0.018	−0.027–0.056
Confidence–Courage–Life satisfaction	0.002	−0.009–0.008	−0.025	−0.070–0.019	−0.006	−0.052–0.037
Readiness–Courage–Flourishing	0.006	−0.003–0.019	0.037	0.025–0.051	0.041	0.029–0.054
Concern–Courage–Flourishing	0.003	−0.010–0.012	0.036	0.006–0.066	0.040	0.013–0.072
Control–Courage–Flourishing	0.003	0.001–0.008	0.038	−0.004–0.076	0.047	0.005–0.083
Curiosity–Courage–Flourishing	0.009	0.001–0.022	−0.010	−0.004–0.026	−0.009	−0.043–0.026
Confidence–Courage–Flourishing	0.013	0.004–0.029	0.041	0.003–0.082	0.054	0.015–0.095

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
