# Peer review of "Courage, Career Adaptability, and Readiness as Resources to Improve Well-Being during the University-to-Work Transition in Italy"

_ijerph, 2021, doi:10.3390/ijerph18062919_

Round 1

Reviewer 1 Report

The study aimed to examine the role of courage, career adaptability and professional readiness towards life satisfaction during university to work transition. However, the study leaves the reader more questions than answers. There are many major concerns that the authors should address:

  • The English language needs attention, such as grammatical mistakes, sentence structures etc. Suggest sending the manuscript for English proofreading. It is hard to correct it one-by-one.
  • Title: Population is from Italy, suggest adding Italy university in the title.
  • Title: Why approaching university degree? It should be “university-to-work transition. This is rather confusing for the readers.
  • Abstract: The study is to analyze? Suggest examining.
  • Abstract: Age ranged from 21 – 49? Suggest only use undergraduate students aged from 21 – 25 years. Adults normally have different challenges.
  • Introduction: Need more explanation regarding the conceptual framework. Seemed like only career transitioning. Little on courage, career adaptability, and professional readiness?
  • The study keeps on repeating and repeating. I read the aim three times repeating.
  • Please delete the hypothesis. Seemed like copying from the thesis only.
  • Please include sample size calculation.
  • Measure: What is the reliability and validity of the questionnaires in the Italian language? Only translate the Career-adapt-abilities Scale? What is the reliability for Courage Measure?
  • Are the questionnaires in the Italian language? Please state the translation process and validity of each questionnaire in the Italian version.
  • Where is the demographic information of the participants? What is the gender differences? Please include a descriptive statistic of demographic profiling.
  • For discussion, please discuss on the results part-by-part. Current form is only descriptive and not discussion.
  • Please separate the limitation and conclusion.
  • The conclusion is too long. Please reduce your conclusion to two paragraphs maximum. It is a conclusion.
  • There are many unnecessary references, please delete it. Seemed like all authors self-cited themselves, which many papers are not in the text.

Reviewer 2 Report

Overall:

The study is interesting, and the paper’s overall goal is clear. This paper’s strengths lie in its methods and results sections: The research design and data analysis were appropriate to the study; the study’s results are well interpreted, and include understandable tables. Below are suggestions that could help improve the paper’s quality.

Abstract:

The findings need to provide more specific details. Upon reviewing the results, two out of five hypotheses (i.e., H9, H15) measuring mediating effects were confirmed. In addition, three out of ten hypotheses (e.g., H1, H2, H14) revealed significant direct influence. Since more than half of the hypotheses were rejected, the authors are encouraged to highlight those hypotheses that yielded significant results.

Introduction:

The authors made a useful introduction of the “protective” factors (e.g., professional readiness, career adaptability, and courage). The protective factors have some overlapping meaning (many have involved the motivation, coping capacity, or adaptive response). Thus, information on the similarities and differences between and among these psychological factors is needed. Furthermore, their contributions to university-to-work transition can be strengthened. Such discussion is important to justify the inclusion of protective factors.

Literature review:

The authors did well to include a fair literature review on the protective factors; however, more convincing justifications or a thorough review are needed for the wellbeing indicators, especially for flourishing.

Materials and Methods

The methods section is appropriate; however, a discussion on the appropriateness of combining multiple data sources (e.g., participants from different universities in the north, center, and south of Italy) is needed. Also, further information on the sample distribution is needed (e.g., age). For example, will participants of different ages respond differently to the survey questions? This concern was raised since the survey samples have an expanded age range (aged from 21 to 49 years).

Results

The results are not correctly worded, which may confuse the readers. It is suggested that the authors conduct further proofreading in this section. For example,

Line 317:

The Results section was numbered incorrectly. I suspect this section should be number five instead of number 3. This has impacted the numbers of headings in the following sections.

Line 380-385:

The authors mentioned that “control and curiosity have not any direct effect on life and flourishing, rejecting Hypothesis 7, 8, 10, 11; thus, the relationship between control and subjective well-being is fully mediated by courage, confirming Hypothesis 9.”

This quoted sentence needs proofed for proper sentence structure (e.g., watch for the adverb used).

Line 388-391:

The authors indicated that “Confidence has not any direct effect, either indirect effect on life satisfaction, so Hypothesis 13 is rejected”

I am confused with this statement. This quoted sentence needs proofed for proper sentence structure (e.g., watch for conjunctions and linking words).

Discussion

The information presented in this section is muddled and lacks a convincing discussion. For example, the discussion does not follow the order of the hypothesis, nor those hypotheses that yielded significant results (or those did not). The discussion for hypotheses 1 and 2 is lacking or limited. In addition, the dimensions of control and confidence discussion are weak. The authors used the exact sentence to explain mediation effects of courage on control and confidence (see 424-425; 427-429). The authors are encouraged to restructure this section.

Another concern lies with the lack of theoretical implications (it is not clear if the discussion can be a meaningful addition to the literature). The authors can incorporate insights founded within the Life Design paradigm (it was briefly mentionedin the literature review section) or other career related theories to strengthen their discussion of the theoretical contributions.

Writing Style: To correct grammatical errors or sentence structure issues, further proofreading is highly recommended. For example,

Line 47-48

This paper contains several long sentences and lacks clarity which may cause the readers lose the focus. Clarity can be improved by restructuring these sentences for example:

The authors mentioned that “the instability that characterizes world of work requires to the individuals to be able to respond adaptively to the continuous changes.” It should read  “requires the individuals to respond…”

Line 53-55: “In particular, we refer in this study to the use of psychological resources that can help the students in approaching the graduation, enabling them to face, in a successful way, the university-to-work transition and the future ones, ensuring their general and domain-specific well-being.”

Line 419-422: “That is, more information about the world is perhaps not perceived as sufficient and functional to achieve the full expression of one's talents and potential and is not able to produce at the same time well-being also on the emotional side such as to counteract the associated uncertainty and fear for the future. “

Line 404: The author(s) wrote “Control and curiosity have not any direct effect on the two indicators of well-being…”

This quoted sentence needs proofed for proper sentence structure (e.g., ordinary verb vs. auxiliary verb).

Reviewer 3 Report

See attached file.

Round 2

Reviewer 1 Report

I suggest accepting the manuscript as the authors had made substantial changes to satisfaction. 

Author Response

Thank you very much for your feedback. 

Reviewer 2 Report

Thank you for your response and the commendable revision. The added information provides a stronger introduction to your research. Nevertheless, I still have some concerns about the following: 1. Literature Review The authors provided a thorough review on the protective variables. A stronger balance is recommended to transition from general overview of the variable to specific college student related articles that focus on a particular variable. For example, a good amount of professional readiness articles was provided; however, college student career readiness related articles are under-represented. The authors are also encouraged to revisit the literature review section to address the association between each pair of the variables highlighted in each hypothesis. This discussion is needed as the SEM highly focuses on the causal relationships between variables. 2. Methods The hypotheses do not read statistically accurate to state X on one side and Y on the other side. Therefore, rewording this section is suggested. Consider adding a figure of framework may be helpful to omit the worded hypotheses. 3. Discussion and Conclusions I commend the great improvement the authors made on this revision. All findings were explained and clearly discussed. To further strengthen the Discussion and Conclusion section, I suggest the authors place additional emphasis on theoretical and practical implications as such information was not clearly presented in the current version. 4. The authors are recommended to do a final proofread before submitting their paper. Make sure their paper is free of grammatical errors, sentence structure issues, and is kept to the journal standard. For example, it is rare to say “...thanks to…” (line 118) in academic writing.

Author Response

  1. Following the reviewer's suggestions we have better focused the literature review on career transition readiness.
  2. We have rephrased the hypothesis
  3. We have strengthened the conclusions
  4. We have further checked the language.